# Dental Students’ Oral Health-Related Quality of Life and Temporomandibular Dysfunction-Self-Rating versus Clinical Assessment

**DOI:** 10.3390/healthcare9101348

**Published:** 2021-10-11

**Authors:** Dagmar Schnabl, Philipp Sandbichler, Maximilian Neumaier, Johannes Girstmair, Fabian Barbieri, Ines Kapferer-Seebacher, René Steiner, Johannes Laimer, Ingrid Grunert

**Affiliations:** 1Department of Operative and Prosthetic Dentistry, Medical University Innsbruck, 6020 Innsbruck, Austria; philipp.sandbichler@tirol-kliniken.at (P.S.); zahnaerzte@dr-jakob-partner.de (M.N.); ordination.girstmair@aon.at (J.G.); ines.kapferer@i-med.ac.at (I.K.-S.); rene.steiner@tirol-kliniken.at (R.S.); info@proimplant.at (J.L.); ingrid.grunert@i-med.ac.at (I.G.); 2Department of Internal Medicine III, Cardiology and Angiology, Medical University of Innsbruck, 6020 Innsbruck, Austria; fabian.barbieri@tirol-kliniken.at

**Keywords:** oral health impact profile, oral health-related quality of life, temporomandibular disorder, dental students, medical students’ disease

## Abstract

The aim of this study was to compare dental students’ self-perception of oral health with the results of a clinical examination of the masticatory system. Seventy-four dental students (38 (51.4%) females and 36 (48.6%) males) completed the Oral Health Impact Profile questionnaire (OHIP-G-14) and underwent a clinical examination according to the Graz Dysfunction Index (GDI). Data were analyzed with descriptive and comparative statistics. Median OHIP-G-14 scores were 3 (IQR 0–6) in the total collective, 4 (1–11) in females, and 2 (0–4) in males (*p* = 0.072). A score of 0 was found in 29.7% of the sample. The results of the GDI were 50% “normal function”, 43.2% “adaptation”, 5.4% “compensation”, and 1.4% “dysfunction”. The comparison of OHIP-G-14 scores and DGI groups showed a significant difference (*p* = 0.031). Based on the questionnaire, less than one third of the sample indicated maximum oral health-related quality of life. In contrast, the GDI revealed “normal function” or “adaptation” in 93.2%. Dental students underappreciated their oral health condition. Health assessments should not be solely questionnaire-based, especially in health professionals (-to-be). To establish a valid diagnosis of the state of health, self-assessment must be complemented by an objective clinical examination, e.g., GDI.

## 1. Introduction

Several questionnaire-based studies have assessed a high prevalence of psychological stress in dental students worldwide [1,2,3,4]. Performance pressure, workload, and self-efficacy beliefs have been identified as the students’ main concerns [2]. The severity of stress tends to increase over the course of studies, corresponding to the transition from the preclinical to the clinical phase of training [3,4]. In dental students, as in the general population, psychological stress is associated with the development of temporomandibular dysfunction (TMD) [5,6,7]. In the prevalence of TMD, female predominance has frequently been assessed [5,7,8,9]. TMD negatively affects oral health-related quality of life (OHRLQoL), as shown in several studies using either the Oral Health Impact Profile (OHIP) or other questionnaires [10,11].

The OHIP is a widely used, validated OHRQoL self-assessment tool that comprises various aspects of oral health such as functional limitation, pain, psychological discomfort, disability, and handicap [12]. Some recent studies have, in different population samples, correlated OHIP-scores and the prevalence of TMD. An Australian national study in 4133 adults observed significantly higher OHIP-14 scores, or lower OHRQoL, respectively, in adults with self-reported TMD (according to the TMD Diagnostic Criteria Questionnaire) than in individuals not reporting TMD experience [13]. Two studies conducted in Asia found that painful TMDs (based on the TMD Diagnostic Criteria Questionnaire) were associated with poorer OHRQoL in young adults [14,15]. Onoda et al. assessed significantly higher OHIP-54 scores in Japanese patients clinically diagnosed with TMD compared to a control group without TMD [16]. A Swedish longitudinal study in preterm-born adolescents found significantly higher mean OHIP-14 scores in participants with self-reported TMD pain than in those without TMD pain [17]. An investigation in 480 Turkish dentistry students revealed higher scores of both the Fonseca-TMD and OHIP-14 questionnaires among senior students as compared to students in earlier years of the study and emphasized the implementation of measures in the dental educational system to raise students’ quality of life [18].

The question of interest in the present study was if the results yielded by the self-assessment tool OHIP-G-14 would reflect the (thoroughly and objectively assessed, not self-reported/questionnaire-based as is the case in most studies mentioned above) state of temporomandibular function/dysfunction in a sample of Austrian (senior) dental students. Due to their preoccupation with disease and dysfunction, dental students constitute a special collective that may present a certain hypersensitivity to bodily sensations and signs. On the other hand, a high level of health consciousness and knowledge on prophylaxis/oral care should result in exceptionally good oral health in this group of young adults. The aim of this clinical study was to correlate dental students’ self-perception of OHRQoL (by use of the OHIP questionnaire) with the GDI (assessed by one clinical examiner). The null hypothesis was that within the score ranges given by each assessment method, OHIP-scores would correspond with equivalent GDI-scores.

## 2. Material and Methods

### 2.1. Ethics Approval and Trial Registration

The Ethics Committee of the Medical University of Innsbruck, Austria, approved the study prospectively (ID 1071/2018). The study was conducted in accordance with the 1964 Helsinki declaration and its later amendments. All participants signed an informed written consent prior to the study enrollment.

This study was registered at the Coordination Center of Clinical Studies of the Medical University of Innsbruck (registration ID 1071/2018).

### 2.2. Subjects

A total of 102 dental students at the Medical University of Innsbruck who were enrolled in the clinical phase of education (7th to 12th semester) were invited to participate as probands in this clinical trial. According to a clinical and radiological assessment that had been performed previously in the course of training, all students had a natural dentition with or without fixed prosthetic restorations. Neither student suffered periodontal disease or deep caries (implicating odontogenic pain). Exclusion criteria were pregnancy, breastfeeding, orthodontic treatment in progress, and orthopedic impairment. Recruitment was accomplished from 4 June to 11 June 2018.

### 2.3. Clinical Parameters and Data Acquisition

Data collection was carried out from 11 June to 29 June 2018. The acquisition of the OHIP-G-14 was performed by one examiner (M.N.) and the clinical examination according to the GDI was carried out by another investigator (P.S.). Each examiner was blinded to the other examiner’s results.

The OHIP is a well-established, validated OHRQoL self-assessment tool that is available in different versions. The original version consists of 49 questions with regard to seven negative aspects of oral health, referring to a certain period of time (e.g., lifetime, preceding year or month) [12]. The grouping into seven domains, (1) functional limitation, (2) physical pain, (3) psychological discomfort, (4) physical disability, (5) psychological disability, (6) social disability, and (7) handicap, was based on a conceptual model of oral health, which uses the framework of the World Health Organization International Classification of Impairments, Disabilities and Handicaps [19]. For settings that demand a succinct assessment, shortened English and German versions containing 14 questions (OHIP-14; OHIP-G-14) regarding the same seven domains were derived [20,21]. For each question, subjects are asked how frequently they have experienced the impact (0 = never, 1 = hardly ever, 2 = occasionally, 3 = fairly often, 4 = very often). For each coded response, different weighting factors (reflecting the populations’ judgements about the relative unpleasantness of each impact) were developed for different OHIP versions and different populations [20]. For the OHIP-G-14, reference values (assessed in the German general population) are available [22].

In this study, the OHIP-G-14 was anonymously assessed with the questionnaire developed by John et al. [21,22], which refers to the English version by Slade et al. 1997 [20]. Regarding seven aspects of oral health, two questions each were answered by use of scores 0 to 4. No weighting factors were used. Thus, the range of possible OHIP summation scores was 0 (maximum OHRQoL) to 56 (severest oral health impairment). Reference period was the preceding month.

The Graz Dysfunction Index (GDI) presents a comprehensive clinical examination for the assessment of the functional state of the stomatognathic system and is based on the clinical function analysis [23]. It comprises 38 parameters assigned to six domains, (1) anamnesis and inspection (targeting signs of parafunction), (2) pain (with respect to its localization, radiation, characteristics, duration, and intensity), (3) joint mobility (including the extent and quality of opening, pro-/retrusion, and side shift), (4) joint noise (e.g., clicking with/without reduction or crepitation), (5) occlusion (including Angle class, deep or open bite, crossbite, nonocclusion, and gliding), and (6) muscle pain on palpation (with regard to its localization and intensity). Altogether, the GDI aims a qualitative and quantitative rating of anamnestic parameters and symptoms. Summation scores are assigned to four classes (normal function, adaptation, compensation, and dysfunction), which categorize the severity of temporomandibular dysfunction.

In the present study, the assessment of the GDI was performed in a separate session by one examiner. Intra-examiner reliability was assessed by use of the intra-class correlation coefficient (ICC), defined as ICC = r × 2/[s.d._1_/s.d._2_ + s.d._2_/s.d._1_ + (M_1_ − M_2_)^2^/s.d._1_ × s.d._2_] (r = Pearson correlation coefficient; M_1_/M_2_ and s.d._1_/s.d._2_ = mean and standard deviation of measurements assessed at sessions I and II) [24,25]. Values of ICC ≤ 0.75 were defined as moderate to poor, those >0.75 and <0.90 as good, and those ≥0.90 as high. In each individual, 38 parameters assigned to the six domains mentioned above were assessed and tabularized. Summation scores <15 were classified as “normal function”, 15–35 as “adaptation”, 36–65 as “compensation”, and >65 as “dysfunction” [23].

### 2.4. Statistical Analysis 

Standard descriptive analysis was used to summarize the data. Qualitative variables are reported as number and percentage, quantitative variables as median and interquartile range. Distribution of continuous variables was determined by using the Kolmogorov–Smirnov test. According to the distribution, either an independent t-test or Mann–Whitney *U* test were used to assess statistical significance in differences. Metric variables were also converted into categorical variables by using predefined cut-off values. Group-specific differences for categorical variables were compared by using the Pearson Chi-Square test. Categorical variables were also compared by use of Kruskal–Wallis test. Statistical analysis was conducted by using IBM SPSS version 21 (IBM Corporation, Armonk, NY, USA). *p*-values ≤ 0.05 were regarded as statistically significant.

## 3. Results

### 3.1. Sample Sharacteristics

A total of 74 subjects (38 females (51.4%) and 36 males (48.6%)) volunteered to participate in the study.

### 3.2. Oral Health Impact Profile-German-14

Median (interquartile range) OHIP-G-14 score amounted to 3 (0–6) in the total collective, 4 (1–11) in females, and 2 (0–4) in males. The difference in OHIP scores was not significant between females and males (*p* = 0.07). Figure 1 illustrates the distribution of OHIP-G-14 scores in the study sample. Twenty-two participants (29.7%) (nine females (23.6%) and thirteen males (36.1%)) had an OHIP score of 0.

### 3.3. Graz Dysfunction Index

To assess intra-examiner reliability, duplicate evaluations were performed in ten subjects, from which the measurement agreement was calculated by reliability analysis. The ICC for the scoring of the GDI was high (ICC = 0.99).

The distribution of DGI scores is illustrated in Figure 2. The median (interquartile range) was 14.5 (7–23) in the total sample, 15.5 (7.8–24.3) in females, and 14 (6.3–21) in males (*p* = 0.40). Table 1 shows the distribution of GDI scores in the six subscales by gender. In the pain category, inputs of 0 (0–0) for both males and females were assessed. Tendentially, females suffered pain more frequently than men (*p* = 0.09). Fifty percent of the sample were assigned to the “normal function” group, 43.2% to the “adaptation” group, 5.4% to the “compensation” group, and 1.4% to the “dysfunction” group. Table 2 displays the distribution of dysfunction groups by gender.

The assessment of differences in OHIP between GDI groups is shown in Table 3. Mid-ranges amounted to 30.26, 44.52, 44.13, and 54.50 for the dysfunction groups “normal function”, “adaptation”, “compensation”, and “dysfunction”, respectively (Kruskal–Wallis test).

## 4. Discussion

### 4.1. Main Results and Comparison with Other Studies

The present study was set up to compare self-assessed OHRQoL and clinically assessed TMD function in a sample of students with a raised awareness of oral health issues. For the German general population, John et al. provided reference data for the German versions of the OHIP questionnaire by means of a survey in 2050 probands aged 16 to 79 years [22]. In that survey, half of the dentate subjects without removable prostheses (n = 1541) had an OHIP-G-14 summation score of 0, reflecting maximum OHRQoL, and 90% had OHIP-G-14 scores ≤11, reflecting minor impairment (as compared to a maximum possible score of 56). Median OHIP-G-14 score was 0 in the sample of dentate probands. In the present study, in which, conformingly, no weighting factors were used, median OHIP-G-14 score amounted to 3. A summation score of 0 was present in only 29.7% of the probands, and OHIP-G-14 scores ≤11 were assigned to 89.2%. The subjectively perceived presence of maximum OHRQoL in less than one third of the study sample seems remarkable with respect to the probands’ young age. Although assessment of exact age was not permitted by the local ethics committee in order to preserve the participants’ anonymity, it is assured that at the time of the study enrollment, most probands were in their mid-twenties and a few were in their early thirties. While (according to the OHIP-G-14 results) barely one third of the study collective indicated complete absence of any oral health impairment, the clinical examination (GDI) revealed “normal function” (largely absence of symptoms) in 50% of the study collective and “adaptation” (scarce or slight symptoms, treatable by a reduction in stressors or behavior modification measures [23]) in 43.2%. Thus, the null hypothesis was rejected. “Compensation” (presence of major symptoms implicating structural changes [23]) was found in 5.4%, and “decompensation” (reflecting severe impairment associated with structural changes [23]) was assessed in 1.4%.

For comparison, an epidemiologic study in a randomly selected sample of Swedish probands found a prevalence of reported frequent TMD symptoms in 13% by use of a questionnaire for signs and symptoms of TMD, whereas only 3% were classified as having severe or moderate clinical signs of dysfunction by use of a clinical dysfunction index at the age of 35 years [26]. In that study, in accordance with several other studies [5,7,8,9], women reported TMD symptoms significantly more often than men. In the present study, women had tendentially higher OHIP-G-14 and DGI scores, but differences in gender were not statistically significant. This conforms to the findings by Helkimo, who assessed roughly the same prevalence of TMD among females and males in a sample of 321 Lapps (aged 19 to 65 years) in Northern Finland by use of both an anamnestic and a clinical dysfunction index [27]. Only a few gender differences were found in that investigation: women had a significantly higher frequency of headache, pain in the neck and shoulders, and fatigue of the jaws.

A study in more than 2000 Japanese university students aimed to elucidate the associations among OHRQoL and clinical oral health status [28]. Mean OHIP-14 score was 1.92 ± 5.47. In non-dentistry students, OHRQoL was associated with oral pain, the decayed missing filled teeth score, malocclusion, recurrent apthous stomatitis, and subjective symptoms of TMD. A cross-sectional investigation in 244 South-East Asian polytechnic students (207 females, 37 males) revealed an absence of TMD in 58.2%, mild TMD in 32.4%, and moderate TMD in 9.4% by use of Fonseca’s Anamnestic Index questionnaire, and a significant association between the severity of TMD and QoL and OHIP subscores, respectively [29]. A recent Turkish study in dentistry students assessed mean OHIP-14 scores as 5.83 ± 7.14 in females and 4.64 ± 6.52 in males (*p* > 0.05). According to the Fonseca Anamnestic Index, 46.6% did not have TMD, 46% had mild TMD, 4.6% moderate, and 2.7% severe TMD. A positive, statistically significant relationship was present between the Fonseca TMD scores and mean OHIP-14 scores. Prevalence of TMD was higher among senior students as compared to junior students, which might be explained by the increase in stress and overloading towards the end of the studies [18]. Compared to that study, the dental students investigated in the present study reported far better OHRQoL.

### 4.2. Advantages and Disadvantages of the Study

While the internationally well-established Research Diagnostic Criteria for Temporomandibular Disorders and their amendment, the Diagnostic Criteria for Temporomandibular Disorders, have prevailed in the consistent categorization of TMD patients with respect to the affected joint structures [30,31], they might result in false negative diagnoses in individuals with compensated TMD or only mild TMD symptoms [32]. The GDI was used as assessment tool in the present study, as (by analogy with Helkimo’s index [33]) it is geared to also detect subclinical TMD manifestations and allows a classification of the severeness of TMD [32]. In contrast to most of the previous studies discussed above, in the diagnosis of function/dysfunction of the masticatory system, the present investigation was based on a substantial clinical examination rather than questionnaire assessment. The examination of all probands by one and the same experienced examiner ensured reliable results, as confirmed by the favorable ICC.

One main disadvantage of this investigation is the lack of a control group. Regarding the (rather small) convenience sample, a selection bias in terms of preference of exceedingly eager or impaired volunteers must be considered. A further limitation of this study is accounted for by the relatively short OHIP-G-14 questionnaire reference period of one month. Nevertheless, the use of the preceding month as reference certainly provided a snapshot of the self-perceived state of oral health within a demanding phase of the dental studies. Another limitation is the different approach to the assessment of the oral health state/temporomandibular function between the questionnaire OHIP-G-14 and the clinical examination GDI by use of non-identical subgroups. All the same, both methods largely cover the spectrum of relevant findings related to temporomandibular (dys)function and oral health (impairment) and provide substantial data using the respective scores. Alas, mucosal moistening disorders, which may cause impairment of OHRQoL, are not taken into account in the GDI. However, salivary gland dysfunction should not be a common problem in the investigated age group.

### 4.3. Clinical Relevance

A great number of questionnaire studies have found a considerable prevalence of psychological distress frequently associated with TMD amongst dental and medical students [1,2,3,4,34,35]. Aside from personal factors (such as irritability, diffidence, or depression) and environmental influences (including competition), an extensive workload, high self-efficacy, and performance pressure may be the reasons for the increased prevalence of (perceived) oral health impairment in students receiving medical education [2]. The present study in dental students found a discrepancy between questionnaire-assessed self-perception and the results of a thorough clinical examination. While the results of the OHIP-G-14-questionnaire suggest a relatively high prevalence of oral health impairment in the sample as compared to the general population, the results of the GDI revealed normal or adapted temporomandibular function in 93% and compensated function in 5.4% of the study collective. Only 1.4% (one female proband) presented “decompensation”. Thus, dental students apparently underappreciated their oral health condition. This may be accounted for by a certain hypersensitivity to or aggravation of constitutional sensations and a conviction of disease resulting from their preoccupation with the bodily health, by analogy with medical students [36]. In recent years, the phenomenon “medical students’ disease” has been conceptualized as a normal process based on perceptual–cognitive and emotional components rather than a form of hypochondriasis [37]. Intriguingly, in a recent study, no difference in health anxiety and hypochondriac/help-seeking behavior was assessed between medical and non-medical students [38]. In dependence of the perceived psychological stress, dental students should implement coping strategies such as task or time schedule optimization, lifestyle modification (e.g., increase in recreation, exercise, or social life), or the improvement of learning conditions (e.g., formation of learning communities or use of alternative learning strategies). In cases of severe distress, psychological counseling might be indicated. In the context of overloading, our study may even prompt a reconsideration of education policies concerning the conditions of studying dentistry.

## 5. Conclusions

Our findings imply that, in the diagnosis of TMD, dental professionals should not rely on sole questionnaire assessment, particularly in dental or medical students or colleagues. Oral health studies should not be based on questionnaire assessment alone, as is often the case. In general, in the evaluation of the state of oral health and temporomandibular function, self-assessment needs to be complemented by an objective clinical examination in order to establish a valid diagnosis of the condition of the stomatognathic system. To this effect, the GDI may serve as an appropriate tool. Therefore, this study comparing subjective and objective oral health findings might be seen as a pilot scheme to investigate also other population groups, preferably in a case–control setting.

## Figures and Tables

**Figure 1 healthcare-09-01348-f001:**
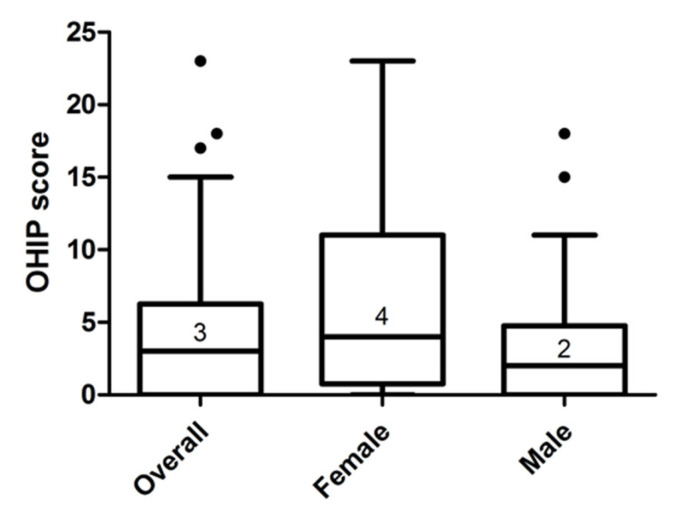
Distribution of OHIP-G-14 scores in the study sample (n = 74). Median (interquartile range): 3 (0–6) in the total sample; 4 (1–11) in females; 2 (0–4) in males; *p* = 0.07; Mann–Whitney *U* test.

**Figure 2 healthcare-09-01348-f002:**
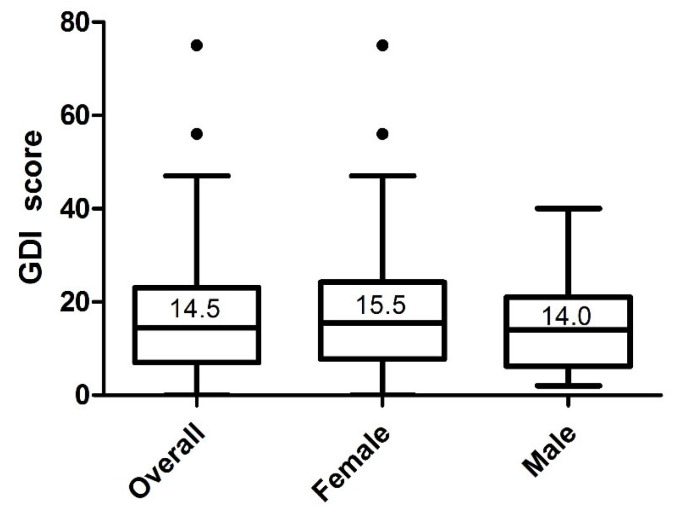
Distribution of the Graz Dysfunction Index scores in the study sample (n = 74). Median (interquartile range): 14.5 (7–23) in the total sample; 15.5 (7.8–24.3) in females; 14.0 (6.3–21) in males; *p*-value = 0.40; Mann–Whitney *U*-test.

**Table 1 healthcare-09-01348-t001:** Median (interquartile range) of Graz Dysfunction Index scores in subscales, by gender.

	Graz Dysfunction Index Score		
Gender	Females (n = 38)	Males (n = 36)	*p*-Value *
Anamnesis and inspection	4 (4–6)	4 (2.5–8)	0.93
Pain	0 (0–0)	0 (0–0)	0.09
Joint mobility	4 (0–7)	3 (0–6)	0.42
Joint noise	0.5 (0–8)	0 (0–8.5)	0.68
Occlusion	1.5 (0–5)	1 (0–3)	0.78
Muscle pain on palpation	0 (0–1.25)	0 (0–0)	0.22

n, number * Mann–Whitney *U* test.

**Table 2 healthcare-09-01348-t002:** Distribution of dysfunction groups according to the Graz Dysfunction Index, by gender.

	Dysfunction Group				Total
	Normal Function	Adaptation	CompenSation	DysFunction	
Males, n (%)	21 (58.3)	14 (38.9)	1 (2.8)	0 (0)	36 (100)
Females, n (%)	16 (42.1)	18 (47.4)	3 (7.9)	1 (2.6)	38 (100)
Total, n (%)	37 (50.0)	32 (43.2)	4 (5.4)	1 (1.4)	74 (100)

n, number; %, percent. *p*-value = 0.07; Pearson Chi-Square test.

**Table 3 healthcare-09-01348-t003:** Comparison of dysfunction groups (GDI) and OHIP-G14 scores.

Dysfunction Group (GDI)	OHIP Median (IQR)
Normal function (n = 37)	1 (0–4)
Adaptation (n = 32)	4 (1.25–10.75)
Compensation (n = 4)	4.5 (17.5–5.75)
Dysfunction (n = 1)	6 (6–6)

n, number. *p*-value = 0.031; Kruskal–Wallis test.

## Data Availability

All data generated or analyzed during this study are included in this published article.

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
