# Peer review of "Dental Students’ Oral Health-Related Quality of Life and Temporomandibular Dysfunction-Self-Rating versus Clinical Assessment"

_healthcare, 2021, doi:10.3390/healthcare9101348_

Round 1

Reviewer 1 Report

Dear Authors, first of all I would like to congratulate You on your work. The introduction provides a good, generalized background of the topic.   Introduction- this section is too long and should be more focused on the topic in question.

Discussion- this paragraph should be rearranged. It is very chaotic. Please do not repeat information from Introduction and try to be more focused. Rewrite this section using following paragraphs: main results and clinical relevance; comparison with other studies; advantages and disadvantages of the study; conclusions and suggestions for future studies.

Author Response

Comment 1: Dear Authors, first of all I would like to congratulate You on your work. The introduction provides a good, generalized background of the topic.   Introduction- this section is too long and should be more focused on the topic in question.

Response: Dear reviewer, thank you for your favorable comments. The introduction was shortened and amended with previous studies so as to clearly explain the goal of this study.

Revised text (pages 1 and 2): OHIP is a widely used validated OHRQoL self-assessment tool that comprises various aspects of oral health such as functional limitation, pain, psychological discomfort, disability, and handicap [12]. Some recent studies have, in different population samples, correlated OHIP-scores and the prevalence of TMD. An Australian national study in 4133 adults observed significantly higher OHIP-14 scores, or lower OHRQoL, respectively, in adults with self-reported TMD (according to the TMD Diagnostic Criteria Questionnaire) than in individuals not reporting TMD experience [13]. Two studies conducted in Asia found that painful TMDs (based on the TMD Diagnostic Criteria Questionnaire) were associated with poorer OHRQoL in young adults [14,15]. Onoda et al. assessed significantly higher OHIP-54 scores in Japanese patients clinically diagnosed with TMD compared to a control group without TMD [16]. A Swedish longitudinal study in preterm-born adolescents found significantly higher mean OHIP-14 scores in participants with self-reported TMD pain than in those without TMD pain [17]. An investigation in 480 Turkish dentistry students revealed higher scores of both the Fonseca-TMD- and OHIP-14-questionnaires among senior students as compared to students in earlier years of the study and emphasized the implementation of measures in the dental educational system to raise students’ quality of life [18].    

The question of interest in the present study was, if the results yielded by the self-assessment tool OHIP-G-14 would reflect the (thoroughly and objectively assessed, not self-reported/questionnaire-based as is the case in most studies mentioned above) state of temporomandibular function/dysfunction in a sample of Austrian (senior) dental students.

Comment 2: Discussion- this paragraph should be rearranged. It is very chaotic. Please do not repeat information from Introduction and try to be more focused. Rewrite this section using following paragraphs: main results and clinical relevance; comparison with other studies; advantages and disadvantages of the study; conclusions and suggestions for future studies.

Response: We revised and rearranged the discussion accordingly.

Reviewer 2 Report

Schnabl et al examined OHIP-G14 and GDI in a cohort of dental student and tried to establish a correlation between them. They concluded that health studies should not be based on sole questionnaire assessment.

In the abstract

  • The first sentence goes like “the aim of this study was to compare…”. However, the real aim was to correlate. This needs to be corrected.
  • It said GDI was performed by one blinded examiner. Later in 2.3, this was changed to two blinded examiners. What do you mean by blinded? These examiners just examined subjects and assigned scores.  What were they blinded to?

In the introduction

  • The authors emphasized on describing detailed information on OHIP (complete and OHIP-14), and GDI. These descriptions should belong to material and methods.
  • One major drawback of this session is that the authors failed to convince me as a reviewer on: Why do we need to correlate OHRQoL with GDI? Are there any similar studies with similar goals? If there is, did they show they correlate or not? Why targeting young dental students instead of a more general population?

In the results

  • Table 1 reported median. In the Pain category, both inputs were zero for females and males; however the P values were 0.09. In this sense, the author should at least indicate which group had relative higher pain levels?
  • Table 3 reporting the correlation between OHIP-G14 and GDI. P values were 0.031, what was Pearson’s r? The authors used all rank test for comparison, thus assumed data distribution was non-normal. In this situation, I recommend conducting spearman correlation as an addition to see the statistics.
  • Is it possible to use OHIP-G14 to predict clinical results by GDI by statistical methods?

The authors need to dig more into the meaning of this research project. It is not convincing at all if this paper concludes that “health studies should not be based on sole questionnaire assessment”.  We all know it is impossible to base any studies solely on questionnaire; examinations need to be always considered. If the conclusion is this, the rationale of the study is undermined.

Author Response

Comment 1: In the abstract The first sentence goes like “the aim of this study was to compare…”. However, the real aim was to correlate. This needs to be corrected.

Response: Although proclaiming the calculation of a correlation, we actually meant the assessment of differences in OHIP between the GDI groups using the Kruskal Wallis test. Thus, “compare” seems correct.

Comment 2: It said GDI was performed by one blinded examiner. Later in 2.3, this was changed to two blinded examiners. What do you mean by blinded? These examiners just examined subjects and assigned scores.  What were they blinded to?

Response: We clarified this issue in the M&M section and removed it from the abstract.

Revised text (M&M, subsection 2.3, page 2): The acquisition of the OHIP-G-14 was performed by one examiner (M.N.) and the clinical examination according to the GDI was carried out by another investigator (P.S.). Each examiner was blinded to the other examiner’s results.

Comment 3: In the introduction The authors emphasized on describing detailed information on OHIP (complete and OHIP-14), and GDI. These descriptions should belong to material and methods.

Response: We shifted the descriptions into the M&M section.

Comment 4: One major drawback of this session is that the authors failed to convince me as a reviewer on: Why do we need to correlate OHRQoL with GDI? Are there any similar studies with similar goals? If there is, did they show they correlate or not? Why targeting young dental students instead of a more general population?

Response: We amended the introduction with previous studies and explained our goal more precisely.

Revised text (pages 1 and 2): OHIP is a widely used validated OHRQoL self-assessment tool that comprises various aspects of oral health such as functional limitation, pain, psychological discomfort, disability, and handicap [12]. Some recent studies have, in different population samples, correlated OHIP-scores and the prevalence of TMD. An Australian national study in 4133 adults observed significantly higher OHIP-14 scores, or lower OHRQoL, respectively, in adults with self-reported TMD (according to the TMD Diagnostic Criteria Questionnaire) than in individuals not reporting TMD experience [13]. Two studies conducted in Asia found that painful TMDs (based on the TMD Diagnostic Criteria Questionnaire) were associated with poorer OHRQoL in young adults [14,15]. Onoda et al. assessed significantly higher OHIP-54 scores in Japanese patients clinically diagnosed with TMD compared to a control group without TMD [16]. A Swedish longitudinal study in preterm-born adolescents found significantly higher mean OHIP-14 scores in participants with self-reported TMD pain than in those without TMD pain [17]. An investigation in 480 Turkish dentistry students revealed higher scores of both the Fonseca-TMD- and OHIP-14-questionnaires among senior students as compared to students in earlier years of the study and emphasized the implementation of measures in the dental educational system to raise students’ quality of life [18].    

The question of interest in the present study was, if the results yielded by the self-assessment tool OHIP-G-14 would reflect the (thoroughly and objectively assessed, not self-reported/questionnaire-based as is the case in most studies mentioned above) state of temporomandibular function/dysfunction in a sample of Austrian (senior) dental students.

Comment 5: In the results Table 1 reported median. In the Pain category, both inputs were zero for females and males; however the P values were 0.09. In this sense, the author should at least indicate which group had relative higher pain levels?

Response: We provided this information in the text.

Revised text (Results, subsection 3.3, page 4): In the pain category, inputs of 0 (0 - 0) for both males and females were assessed. Tendentially, females suffered pain more frequently than men (p = 0.09).

Comment 6: Table 3 reporting the correlation between OHIP-G14 and GDI. P values were 0.031, what was Pearson’s r? The authors used all rank test for comparison, thus assumed data distribution was non-normal. In this situation, I recommend conducting spearman correlation as an addition to see the statistics.

Response: Thank you for pinpointing this important issue. Please see Response to Comment 1.

Revised text (Results, subsection 3.3, page 5): The assessment of differences in OHIP between GDI groups is shown in Table 3. Mid-ranges amounted to 30.26, 44.52, 44.13, and 54.50 for the dysfunction groups “normal function”, “adaptation”, “compensation”, and “dysfunction”, respectively (Kruskal-Wallis test).

Comment 7: Is it possible to use OHIP-G14 to predict clinical results by GDI by statistical methods?

Response: Thank you for your comment. Although we were able to observe higher OHIP values throughout the GDI groups, which was also found to be statistically significant, we are unable to provide clear cutoffs for predicting results of GDI given the small number of patients in the compensation and especially in the dysfunction group. Nonetheless, it seems that higher OHIP results tend to have an increased probability for abnormal results in GDI.

Comment 8: The authors need to dig more into the meaning of this research project. It is not convincing at all if this paper concludes that “health studies should not be based on sole questionnaire assessment”.  We all know it is impossible to base any studies solely on questionnaire; examinations need to be always considered. If the conclusion is this, the rationale of the study is undermined.

Response: Dear reviewer, thank you for your constructive critique and your incitations. We revised the discussion, comparing our findings with those of previous oral health studies in different population samples worldwide (including non-dental and dental students), many of which are based on sole questionnaire assessment. We hope that you will find our conclusions consistent and deem our manuscript suitable for publication.

Reviewer 3 Report

I think it is a meaningful study to examine the relationship between the qualitative evaluation of dental students' oral health-related quality of life and the GDI. However, the explanation of the necessity of the research is insufficient, and a review is necessary as a scientific research in the research method.

(Materials and methods) A total of 102 dental students were recruited for the study, and it is recommended that the final analysis target be 74 students. In addition, the rationale for determining the appropriate sample size should be explained.

(Results) Table 3 shows the results of correlation analysis between GDI and OHIP-14. However, the analysis method using the Kruskal-Wallis test is not suitable to analyze the correlation with the difference analysis of the median OHIP-14 for each subgroup of GDI. We recommend that you try another suitable analysis method that can analyze the correlation between two variables as continuous variables.

(Discussion) The limitations of this study are poorly described. In addition, there is a need for suggestions for the significance of this study and future research.

Author Response

Comment 1: I think it is a meaningful study to examine the relationship between the qualitative evaluation of dental students' oral health-related quality of life and the GDI. However, the explanation of the necessity of the research is insufficient, and a review is necessary as a scientific research in the research method.

Response: Thank you for your favorable comments. The introduction was shortened and amended with previous studies so as to clearly explain the goal of this study.

Revised text (pages 1 and 2): OHIP is a widely used validated OHRQoL self-assessment tool that comprises various aspects of oral health such as functional limitation, pain, psychological discomfort, disability, and handicap [12]. Some recent studies have, in different population samples, correlated OHIP-scores and the prevalence of TMD. An Australian national study in 4133 adults observed significantly higher OHIP-14 scores, or lower OHRQoL, respectively, in adults with self-reported TMD (according to the TMD Diagnostic Criteria Questionnaire) than in individuals not reporting TMD experience [13]. Two studies conducted in Asia found that painful TMDs (based on the TMD Diagnostic Criteria Questionnaire) were associated with poorer OHRQoL in young adults [14,15]. Onoda et al. assessed significantly higher OHIP-54 scores in Japanese patients clinically diagnosed with TMD compared to a control group without TMD [16]. A Swedish longitudinal study in preterm-born adolescents found significantly higher mean OHIP-14 scores in participants with self-reported TMD pain than in those without TMD pain [17]. An investigation in 480 Turkish dentistry students revealed higher scores of both the Fonseca-TMD- and OHIP-14-questionnaires among senior students as compared to students in earlier years of the study and emphasized the implementation of measures in the dental educational system to raise students’ quality of life [18].    

The question of interest in the present study was, if the results yielded by the self-assessment tool OHIP-G-14 would reflect the (thoroughly and objectively assessed, not self-reported/questionnaire-based as is the case in most studies mentioned above) state of temporomandibular function/dysfunction in a sample of Austrian (senior) dental students.

Comment 2: (Materials and methods) A total of 102 dental students were recruited for the study, and it is recommended that the final analysis target be 74 students. In addition, the rationale for determining the appropriate sample size should be explained.

Response: It was a convenience sample. 102 students were invited and 74 volunteered to participate. We clarified this.

Revised text (M&M, subsection 2.2, page 2): 102 dental students of the Medical University of Innsbruck who were enrolled in the clinical phase of education (7th to 12th semester) were invited to participate as probands in this clinical trial.

Revised text (Results, subsection 3.1, page 4): 74 subjects (38 females (51.4%) and 36 males (48.6%)) volunteered to participate in the study.

Comment 3: (Results) Table 3 shows the results of correlation analysis between GDI and OHIP-14. However, the analysis method using the Kruskal-Wallis test is not suitable to analyze the correlation with the difference analysis of the median OHIP-14 for each subgroup of GDI. We recommend that you try another suitable analysis method that can analyze the correlation between two variables as continuous variables.

Response: Thank you for pointing out this error in nomenclature. Although proclaiming the calculation of a correlation, we actually meant the assessment of differences in OHIP between the GDI groups using the Kruskal Wallis test. We have addressed this section accordingly.

Revised text (Results, subsection 3.3, page 5): The assessment of differences in OHIP between GDI groups is shown in Table 3. Mid-ranges amounted to 30.26, 44.52, 44.13, and 54.50 for the dysfunction groups “normal function”, “adaptation”, “compensation”, and “dysfunction”, respectively (Kruskal-Wallis test).

Comment 4: (Discussion) The limitations of this study are poorly described. In addition, there is a need for suggestions for the significance of this study and future research.

Response: Thank you for these incitations. We elaborated advantages/disadvantages and conclusions.

Revised text (Discussion, subsection 4.2): One main disadvantage of this investigation is the lack of a control group. Regarding the (rather small) conveniance sample, a selection bias in terms of preference of exceedingly eager or impaired volunteers must be considered.

Revised text (Conclusions, page 5): Our findings imply that, in the diagnosis of TMD, dental professionals should not rely on sole questionnaire assessment, particularly in dental or medical students or colleagues. Oral health studies should not be based on questionnaire assessment alone, as is often the case. In general, in the evaluation of the state of oral health and temporomandibular function, self-assessment needs to be complemented by an objective clinical examination in order to establish a valid diagnosis of the condition of the stomatognathic system. To this effect, the GDI may serve as an appropriate tool. Therefore, this study comparing subjective and objective oral health findings might be seen as a pilot scheme to investigate also other population groups, preferably in a case-control setting.

Reviewer 4 Report

Dear Authors,

I reported few suggestions to improve your article:

I suggest the Authors briefly explain more in the introduction and also in discussing the reasons to have decided to evaluate Dental students and not students of other courses like Medicine, where clinical training is expected too.

The Authors reported 102 enrolled students, and in the results, only 74 were reported. Please report why the students are exit from the studied group.

In supplementary files, please revised the tables, because it is not clear the meaning of the little boxes reported in right and left and outside right and left column. Furthermore, revised the format of tables in order to check the allineaments among the title and above descriptions.

Minor:

Please make the font of the text uniform and check the space errors (for example after “abstract and keywords”).

Best Regards

Author Response

Comment 1: Dear Authors, I reported few suggestions to improve your article: I suggest the Authors briefly explain more in the introduction and also in discussing the reasons to have decided to evaluate Dental students and not students of other courses like Medicine, where clinical training is expected too.

Response: Dental students constitute a special collective with a raised awareness of oral health. We amended the introduction and the discussion with previous studies in different population groups (including non-dental and dental students), and believe that our investigation may serve as a pilot scheme for further studies in other samples.

Revised text (Introduction, page 2): An investigation in 480 Turkish dentistry students revealed higher scores of both the Fonseca-TMD- and OHIP-14-questionnaires among senior students as compared to students in earlier years of the study and emphasized the implementation of measures in the dental educational system to raise students’ quality of life [18].

Revised text (Discussion, pages 7 and 8): An investigation in 480 Turkish dentistry students revealed higher scores of both the Fonseca-TMD- and OHIP-14-questionnaires among senior students as compared to students in earlier years of the study and emphasized the implementation of measures in the dental educational system to raise students’ quality of life [18].

Comment 2: The Authors reported 102 enrolled students, and in the results, only 74 were reported. Please report why the students are exit from the studied group.

Response: It was a convenience sample. 102 students were invited and 74 volunteered to participate. We clarified this.

Revised text (M&M, subsection 2.2, page 2): 102 dental students of the Medical University of Innsbruck who were enrolled in the clinical phase of education (7th to 12th semester) were invited to participate as probands in this clinical trial.

Revised text (Results, subsection 3.1, page 4): 74 subjects (38 females (51.4%) and 36 males (48.6%)) volunteered to participate in the study.

Comment 3: In supplementary files, please revised the tables, because it is not clear the meaning of the little boxes reported in right and left and outside right and left column. Furthermore, revised the format of tables in order to check the allineaments among the title and above descriptions.

Response: For copyright reasons, supplementary files were removed. Tables were revised.

Comment 4: Minor: Please make the font of the text uniform and check the space errors (for example after “abstract and keywords”).

Response: We apologize. The font problem should be solved through the journal’s layout. We checked the text for space errors.

Dear reviewer, thank you for your constructive critique. We hope that you will find our revisions satisfactory and deem our manuscript suitable for publication.

Round 2

Reviewer 3 Report

The authors have revised manuscript according to the review opinions, so I think it can be published in its current state.

Reviewer 4 Report

Dear Authors,

Thank you for your extensive revision, which I fully agree with.

Best regards.